# Effects of Formula Milk Feeding in Premature Infants: A Systematic Review

**DOI:** 10.3390/children9020150

**Published:** 2022-01-24

**Authors:** Marta Moreira-Monteagudo, Raquel Leirós-Rodríguez, Pilar Marqués-Sánchez

**Affiliations:** 1Faculty of Physical Therapy, Universidade de Vigo, Campus a Xunqueira, s/n, 36005 Pontevedra, Spain; marta_mm_94@hotmail.com; 2SALBIS Research Group, Faculty of Health Sciences, University of León, Astorga Ave. s/n, 24401 Ponferrada, Spain; mpmars@unileon.es

**Keywords:** premature infants, infant formula, feeding methods, breast feeding, intensive care units

## Abstract

The preterm baby is born at a critical period for the growth and development of the gastrointestinal and neuromotor systems. Breast milk is the food of choice for infants during the first months of life, as it provides multiple short- and long-term benefits to preterm and sick newborns. Despite this, breastfeeding is often nutritionally insufficient, requiring the addition of fortifiers. In other cases, it is important to ensure the necessary nutrients and calories, which can be provided by formula milk or pasteurized and fortified donated human milk. However, the specific guidelines for the use of formula milk have not yet been determined. Therefore, a systematic search was considered necessary in order to identify the effects of feeding with formula milk in preterm infants. A systematic search in Scopus, Medline, Pubmed, Cinahl, ClinicalTrials and Web of Science with the terms Infant Formula and Infant Premature was conducted. A total of 18 articles were selected, of which, eight were experimental and ten were observational studies. Among the objectives of the analyzed investigations, we distinguished nine that compared the effects of feeding with formula milk, breast milk and donated human milk, five that evaluated the effects of different compositions of formula milk and/or fortifiers and four investigations that compared the effects of formula milk and donated human milk. In conclusion, when breast milk is insufficient or unavailable, formula milk is a good nutritional option, due to its higher caloric density and protein content. Nevertheless, the preterm infant’s diet should incorporate breast milk to reduce the incidence of morbidities such as necrotizing enterocolitis and sepsis (related to hospital handling of fortifiers and formula milk).

## 1. Introduction

Every year, approximately 15 million babies worldwide are born before term [1]. Prematurity is one of the most important child health problems, since it is the leading cause of perinatal mortality and of 50% of childhood disability, and due to its associated economic and social costs [1,2]. The World Health Organization (WHO) defines a preterm baby as one born before 37 weeks of gestation, and is classified according to gestational age (GA) as: extremely preterm (less than 28 weeks GA), very preterm (28 to 32 weeks), and moderate or late preterm (32 to 37 weeks) [1].

The preterm baby is born at a critical period for the growth and development of the nervous system, and the physiological processes that would take place in utero during the third trimester of gestation are carried out in the neonatal intensive care unit (NICU) [3,4]. Gastrointestinal and neuromotor immaturity and deficits in swallowing–sucking coordination [5] lead to feeding tolerance problems, causing enteral feeding to be delayed or discontinued [6]. One of the consequences of this postponement is the prolongation of parenteral nutrition, which increases the risk of infections, intestinal perforation, necrotizing enterocolitis (NEC), etc. [5,6].

The development of the intestinal microbiota depends on different factors, such as the type of delivery, GA, and feeding [7]. It is known that the composition of the gut microbiota of preterm infants is very poor compared to that of term infants [8], with a lower diversity and higher concentration of pathogenic bacteria and a smaller number of Bifidobacteria and Bacteroides [7]. One of the main nutritional goals in preterm infants is to achieve a weight gain similar to that of the fetus up to 40 weeks’ GA, optimal growth and neurodevelopment, and adequate bone mineralization [9], as a nutritional deficit could have irreparable consequences on both growth and neurological development [3].

Breast milk (BM) is the food of choice for infants during the first months of life, as it provides multiple short- and long-term benefits to preterm and sick newborns [10,11]. This explains why breastfeeding (BF) is a priority in the NICU [10]. Despite this, mothers’ BF is sometimes nutritionally insufficient, leading to the addition of fortifiers [12,13]. In these cases, the World Health Organization guidelines recommends pasteurized and fortified donated human milk (DHM) and, in the absence of it, formula milk (FM) [14]. Some of its advantages are protection against NEC and nosocomial infection, better digestive tolerance, and reductions in healthcare costs [14]. However, in cases where mothers are unable to provide BM, it is justified that FM feeding is necessary [15]. This type of feeding attempts to mimic the properties, composition, and bioavailability of BM, and there is some evidence that FM improves growth [16]. However, the specific guidelines for the use of one or the other type of feeding have not yet been determined [13].

Therefore, a systematic search of scientific publications was considered necessary in order to identify the effects of feeding with FM in preterm infants. As a previous hypotheses, the authors defined that FM may have adverse effects that differ from those of BM and DHM.

## 2. Materials and Methods

### 2.1. Search Strategy

This study was registered on PROSPERO (ID: CRD42021236144) and followed the Preferred Reporting Items for Systematic Reviews and Meta-Analyses (PRISMA) reporting guidelines and the recommendations from the Cochrane Collaboration [17,18]. The PICO question was then chosen as follows: P–population: premature babies; I–intervention: FM fed; C–control: BM- and/or DHM-fed; O–outcome: weight gain, head circumference (HC) and length growth, fecal analysis, gut microbiota analysis, blood tests, and/or intolerance (vomiting and/or diarrhea); S–study designs: experimental and observational studies.

A systematic search of publications was conducted in January 2021 in the following databases: Scopus, Medline, Pubmed, Cinahl, ClinicalTrials.gov, and Web of Science. The search strategy included different combinations with the following Medical Subject Headings (MeSH) terms: Infant Formula and Infant Premature. The search strategy according to the focused PICOS question is presented in Table 1.

### 2.2. Study Selection

After removing duplicates, two reviewers (M.M.-M. and R.L.-R.) independently screened articles for eligibility. In case of disagreement, both reviewers debated until an agreement was reached. For the selection of results, the inclusion criteria established that the articles must have been published in the last five years (from 2015 to the present), FM was administered to the study sample, and that the sample consisted of preterm infants. On the other hand, studies were excluded from this review if weight gain and head growth had a non-experimental or observational methodology (reviews, meta-analyses, editorials…) and their full text was not available.

After screening the data, extracting, obtaining and screening the titles and abstracts for inclusion criteria, the selected abstracts were obtained in full texts. Titles and abstracts lacking sufficient information regarding inclusion criteria were also obtained as full texts. Full text articles were selected in case of compliance with inclusion criteria by the two reviewers using a data extraction form. The two reviewers mentioned independently extracted data from included studies using a customized data extraction table in Microsoft Excel. In case of disagreement, both reviewers debated until an agreement was reached. 

The following data were extracted from the included articles for further analysis: demographic information (title, authors, journal and year), characteristics of the sample (GA, inclusion and exclusion criteria, and number of participants), study-specific parameter (study type, duration of the intervention or retrospective/prospective cohort analysis period, type of feeding provided, quantity of feed, characteristics and origin of the milk administered) and results obtained. Tables were used to describe both the studies’ characteristics and the extracted data. When possible, the results were gathered based on type of intervention applied. The Oxford 2011 Levels of Evidence and the Jadad scale were used to assess the quality of studies.

## 3. Results

### 3.1. Auxological Data

A total of 18 articles were selected, of which eight were experimental studies [19,20,21,22,23,24,25,26] and the remaining ten were observational studies [27,28,29,30,31,32,33,34,35,36] (Figure 1). The characteristics of the applied interventions are presented in Table 2.

Among the objectives of the analyzed investigations, we distinguished nine that compared the effects of feeding with FM, BM, and DHM [21,25,27,28,29,32,33,35,36], five that evaluated the effects of different compositions of FM and/or fortifiers [19,24,26,30,34], and four investigations that compared the effects of FM and DHM [20,22,23,31]. The methodological characteristics of the analyzed studies are detailed in Table 3.

### 3.2. Tolerance and Growth

Brownell et al. [29] identified that for every 10% increase in DHM intake, the rate of weight gain decreased by 0.17 g/kg/day and head circumference (HC) growth also decreased compared to the growth of BM-fed infants. On the other hand, the weight gain increased significantly with increasing FM intake [29,32]. The length growth rate did not show any significant relationship with feeding [29]. There was also a significant association between lower HC and increased DHM, but not with FM [29]. However, Martins-Celini et al. [28] and Lofti et al. [32] did not identify any significant difference in weight or HC between the different feeding modalities at hospital discharge, although the length was significantly shorter in infants fed with BM or DHM compared to the FM group. In Brownell et al. [29], both BM and DHM were fortified (caloric density of 67 kcal per 100 mL) when 100 mL/kg/day was reached and progressed to a total volume of 140–160 mL/kg/day. Martins-Celini et al. [28] did not fortify BF or DHM until the infants reached an intake of 100 mL/kg/day. Cunha et al. [25] have identified that the addition of a multi-nutritional supplement (without further defining its composition) reduced (non-significantly) the incidence of neuropsychomotor development. More specifically, they identified impaired psychomotor development in 33.3% of infants fed exclusively with BM and in 28% of infants fed with BM and supplement [25]. In addition, there were no significant results in the Bayley Scale domains (although the scores were higher in the supplemented group).

Kim et al. [26] compared the growth of preterm infants fed either a standard powder fortifier or a liquid concentrate of extensively hydrolyzed proteins and found that weight and length at one month of age were significantly greater with the liquid fortifier. Moreover, the infants who received the liquid fortifier reached 1800 g significantly earlier than the other group. The HC revealed no statistical differences, with both fortifiers resulting in similar caloric intake and reporting similar incidences of NEC and sepsis. However, it should be noted that significantly fewer children discontinued fortification due to food intolerance in the group consuming the liquid fortifier.

An indirect sign of good feeding tolerance is growth (increase in length, weight, and HC of neonates). In fact, Hogewind-Schoonenboom et al. [34] evaluated the association of the amount of fortified BM or FM with feeding tolerance and growth in preterm infants. Among their results, they identified that residual gastric volumes were significantly lower in the group that received the least amount of BM (as they divided the sample according to the percentage of feeding from the mother and FM). However, there were no significant differences in any tolerance parameter, in the incidence of adverse events or in weight gain, HC, and length. 

Pillai et al. [30] had the specific objective of determining the tolerance to a concentrated liquid fortifier (without further specifying its composition or source). Their results indicated that intolerance occurred in 14% of the infants (of these, 3% suffered sepsis). However, there were no cases of NEC after addition of fortifier. Growth rate increased from 12.5 to 15.9 g/kg/day after addition of fortifier. Baldasarre et al. [19] also aimed to determine the tolerance to an intact protein FM and an extensively hydrolyzed FM. With both options, the time to achieve full enteral feeding was similar, although it was significantly shorter for the group fed intact protein FM. As the achieved feeding volumes increased, greater divergence was observed between the groups in mean enteral intake: at the end of the study, it was significantly higher in the group fed intact protein FM. No significant differences were found in weight, length, HC, tolerance, respiratory status, morbidities or length of hospital stay.

Costa et al. [20] and O’Connor et al. [23] compared the tolerance to both feeding modalities and found that the time to reach enteral feeding was similar in both groups. Although Costa et al. [20] observed that the total protein and calorie intake was significantly higher in the FM-fed group. No significant differences were found in any of the anthropometric variables [20,23] or in cognitive, language, and motor development [23], it was observed that the group fed with FM regained birth weight in a significantly shorter time than those who received DHM [20]. It was also identified that a significantly higher percentage of the DHM-fed group had a cognitive score indicative of neurological impairment [23]. In addition, there was a significantly higher incidence of NEC in the FM-fed group [23].

### 3.3. Microbiota

Cong et al. [36] specifically aimed to analyze the intestinal microbiota and identified that infants who received BM had higher numbers of Clostridiales, Lactobacillales, and Bacillales and smaller numbers of Enterobacteriaceae. In contrast, infants fed DHM and FM had a higher proportion of Enterobacteriaceae. They also identified that α-diversity was significantly higher in the BM group. Regarding ß-diversity, the feeding method was found to be the variable that explained the greatest variance, followed by sex, GA and postnatal age, antibiotic use, and premature rupture of membranes. Along the same lines, Chen et al. [27] aimed to describe differences in the development of intestinal microecology as a function of feeding. They identified that Firmicutes, Proteobacteria, and Actinobacteria accounted for more than 99% of all organisms in the neonatal faeces, and that Bacteroides accounted for 0.3% of the total in all neonatal infants. In short-chain fatty acids (SCFA), propionate was found to be present in lower proportions in both groups and increased later in the BM group; acetic acid and butyrate were the most abundant in both groups (although significantly higher in FM after one week of study and significantly higher in BM after three weeks). In general, all SCFA concentrations were higher in the FM group, with acetic acid being the most abundant. Analyses of fecal DNA samples found that the feeding mode was not associated with significant differences in α-diversity or ß-diversity. It was found that, in BM-fed infants, the gut flora decreased by 30.6% at one month of age. However, in FM-fed infants, it increased on average by 52%. In relation to weight, significant increases were found in both groups, although the increase was significantly greater in the FM group in those infants younger than 32 weeks GA (in those older than 32 weeks the increase was similar). On the first day of the study, there were no significant differences in the microbiota between the two groups, although it was found that, after one month, pentose metabolic pathways, glucuronate interconversions and compound selenium were statistically higher in the BM group, and that there was significantly higher histidine metabolism in the FM group. Finally, Bifidobacteria and Actinomycetes were found to be higher with greater birth weight, Bacteroides increased their proportion with older GA, and Actinomycetes, Pseudomona Aeruginosa, and Burkhol-deria increased with weight gain.

Finally, Jang et al. [35] compared fecal calprotectin levels in infants with and without feeding intolerance (according to absence of vomiting, increased gastric residuals, and abdominal distension). They identified that infants without feeding intolerance had significantly higher GA and birth weight. Hospitalization length and fecal calprotectin levels were significantly higher in infants with feeding intolerance. In turn, the fecal calprotectin level was significantly higher in BM- or FM-fed infants compared to the level found in infants fed with amino acid-based formulas. However, the groups did not differ statistically in their growth rate or weight at discharge.

### 3.4. Long-Term Follow Up (Evolution after Hospital Discharge)

The study with the longest follow-up of the participating infants was that of Toftlund et al. [21]. They specifically aimed to analyze the long-term effects of BM or FM feeding on growth and identified that the FM-fed group achieved faster birth weight recovery and, up to four months, there was significantly faster weight gain (and more so in those infants who were small for their GA). Growth up to six years and growth faltering after 34 weeks showed no significant differences by feeding type. At six years, infants born small for GA had achieved significantly greater gains in weight and length than those of appropriate size; however, they achieved significantly lower weight and length irrespective of feeding type.

In addition, another study conducted a long-term follow-up of the participants and identified that, at one year of age, growth rate, weight, length, and HC were similar in infants fed BM, DHM, and FM [33]. The incidence of metabolic bone disease did not differ by feeding mode [32].

### 3.5. Incidence of Complications

No differences in the incidence of morbidity and mortality [22,31], the use of surfactant or the administration of antibiotics [31] were identified between the groups. In one study, the two modalities did not differ significantly in the time required to reach 120 mL/kg/day or in the duration of parenteral feeding [22]. However, in another study, it was different: the age to reach 50 and 130 mL/kg/day and, consequently, the length of hospitalization was significantly longer in FM-fed infants [31]. 

Willeitner et al. [24] found that a 30 kcal/oz liquid fortifier with a caloric density of 24 kcal/oz did not result in significant improvement in weight, feeding tolerance, caloric intake, sepsis or mortality compared to a standard fortifier (whose composition and caloric density are not specified).

In cases where mechanical ventilation or central venous catheterization was necessary, infants in the FM-fed group required them for a longer period of time [31]. The incidence of death, sepsis, NEC, and bronchopulmonary dysplasia was significantly lower in the DHM group compared to the FM-fed group [31].

## 4. Discussion

The aim of the present investigation was to identify the effects of FM feeding in preterm infants. After analysis of the results were obtained, it could be affirmed that FM is safe and beneficial for preterm infants.

Very low birth weight infants usually present vomiting, increased gastric residuals, and abdominal distension associated with delayed elimination of meconium or bloody stool, which are characteristic signs of enteral feeding tolerance problems [3,6]; this is why this variable has been analyzed in most studies [19,20,22,24,26,30,31,34,35]. The FM had a similar tolerance to human milk (either BM or DHM), except in two studies [26,34]. In one study, a plausible explanation for this finding is the intolerance to casein milk administered in FM [26,37], although, in the other case, no data are provided to justify this finding. Complete enteral feeding was achieved after longer periods with FM [31] or when using liquid fortifier [30] (at 31 and 30 days, respectively) compared to BM. However, the other studies showed that the time did not differ significantly [20,22,34,35], reaching ten days in the case of intact protein FM [19]. Furthermore, in the studies by Costa et al. [20] and Corpeleijn et al. [22], which used both FM and DHM as BM supplementation, the time to reach enteral feeding within the first two weeks of life was the same with both types of feeding. 

Intolerance to enteral nutrition requires more time on parenteral nutrition, which increases the risk of comorbidities and mortality [6]. A higher tendency for NEC, sepsis, and bronchopulmonary dysplasia was found when FM [23,31,32] or a powdered fortifier [24] was used. Although these are two different foods, the finding of complications with both could be due to the fact that in both cases, a powdered substance is administered. In fact, this finding is consistent with previous studies in which a higher rate of NEC was associated with FM intake [38,39] and may be due to bacterial contamination of powdered FM or fortifier during its preparation [40,41]. 

Birth weight recovery occurred earlier in infants fed FM as a supplement to BM [20], exclusively [21] or with a liquid fortifier [26]. These results are contradictory to those obtained previously [41], where BM-fed infants had higher growth than those who received FM. It should be noted that, in one of the studies, as the percentage of DHM intake increased [29] (even though it was fortified), the rate of weight gain decreased. This is possibly due to the different composition of DHM (lower in fat, protein and calories) and the pasteurization process it undergoes [12]. Length was affected with the combination of BM and FM [28], FM [29] and DHM [31]; however, it increased significantly with liquid fortifier [26] and with the combination of FM and BM [32]. 

Psychomotor development was not statistically affected by the feeding method, although the highest scores were found for feeding BM supplemented with FM [23,25]. However, exclusive DHM [23] or BM [25] feeding (both not fortified) resulted in a higher incidence of neurodevelopmental delay (without being statistically significant in the intergroup comparison), possibly due to the lack of supplementation.

BF and its combination with FM resulted in the development of a microbiota with a composition more similar to that of the term infant, with a higher proportion of Firmicutes, Proteobacteria, Actinobacteria, and Bifidobacteria [27,36]. In contrast, not providing BM or DHM feeding implied a higher amount of pathogenic Enterobacteriaceae [36], which could suggest that immunological, nutritional, and microbial properties characteristic of human milk are lost in the process of pasteurization of DHM. However, after one month of life, BM decreases the number of species in the microbiota, whereas FM increases it [27], although not significantly. In any case, BM provides an increasing trend of microbial diversity [27], probably due to prebiotics contained in BM, such as oligosaccharides. The α-diversity was higher with BM [27,36], FM had no significant effect [27], and DHM decreased it [36]. In the latter case, a possible influencing factor would again be the pasteurization process.

Regarding short-term (first weeks of life) and long-term (from six months of age) benefits, studies comparing the effects of feeding during hospital stay [19,20,22,24,26,27,28,29,30,31,32,34,35,36], showed that there were no differences in tolerance or duration of admission regardless of the type of feeding [19,20,22,24,26], except in one of the studies [31]. In this case, FM prolonged the time to reach full enteral feeding and the duration of the hospital stay [31]. In terms of growth, the best results were observed with FM [20,27,31], with the combination of BM and FM [32] and with two liquid fortifiers [26,30]. In the remaining studies, the results were similar [19,24,28,35], and, in one study, DHM was associated with reduced growth [29]. Morbidity did not differ statistically according to the type of feeding [19,22,24,26,34,35], although in several studies there was a higher incidence with FM [30,31,32]. In those studies that assessed long-term effects, growth to 12–18 months was similar regardless of the feeding type [23,25,33], and neuropsychomotor development had higher scores on the Bayley scale with FM [23,25]. 

In studies where calorie density was indicated [19,21,23,24,29,30,31,33], it was observed that providing a higher calorie intake improved growth (weight, length, and HC). This phenomenon was not identified with DHM fortification [23,29]. Since better results were obtained with the feeding of FM, the increased proportion of DHM was even associated with reduced growth [29]. However, when fortifying BM, this negative association was not found [21,25].

Studies comparing different formulations [19,24,26,30,35] have shown that infants who ingested liquid fortifier [24,26,30] had better tolerance [24], recovery, and growth velocity [26,30] than infants who ingested a powder fortifier, even though the addition of a liquid fortifier reduces the proportion of BM. Although in two studies the differences between the two fortifiers in terms of morbidity were not significant, a higher incidence of NEC [24] and a significantly higher percentage of children had to stop fortification due to intolerance [26] to the powdered fortifier. This could be due to the fact that the liquid fortifier is more sterile and the risk of contamination is lower. One of them was hydrolyzed [26], which may have contributed to the better tolerance. However, this statement is not supported by the study of Baldasarre et al. [19], where it was observed that children consuming FM with intact protein achieved full enteral feeding in a shorter time than those on highly hydrolyzed FM. Finally, although only one study used an amino acid-based formula [35], it showed good results in reducing fecal calprotectin in children with feeding intolerance compared to BM or FM, without affecting growth.

Regarding the methodological limitations of this review, it should be noted that many studies are observational, in which the lack of randomization and control group limits the generalizability of the results obtained. It should also be noted that the sample sizes were small, thus the results may not be entirely accurate and extrapolation to the entire population of preterm infants is not possible. However, as strengths, it is worth highlighting that the methods used to measure the different variables are objective. In addition, this work has included a large number of studies, covering all degrees of prematurity and the application of the different types of feeding that are currently available.

Finally, future lines of research would require new randomized clinical trials with larger sample sizes and longer follow-up periods to assess the impact of the neonatal feeding mode in the long term. It is also necessary for future studies to determine what the caloric density and nutritional values of the feedings should be, which would allow for a better comparison of the results. Finally, it would also be interesting to conduct more studies that include liquid fortifiers in their feeding regimen, as the results obtained in the few studies that used them were very positive.

## 5. Conclusions

When BM is insufficient or unavailable, FM is a good nutritional option, due to its higher caloric density and protein content, as it improves growth and psychomotor development. Nevertheless, the preterm infant’s diet should incorporate BM to reduce the incidence of morbidities such as NEC and sepsis (related to hospital handling of fortifiers and FM). 

In addition, further research is needed on the potential benefits on the gut microbiome of the combination of BM and FM.

## Figures and Tables

**Figure 1 children-09-00150-f001:**
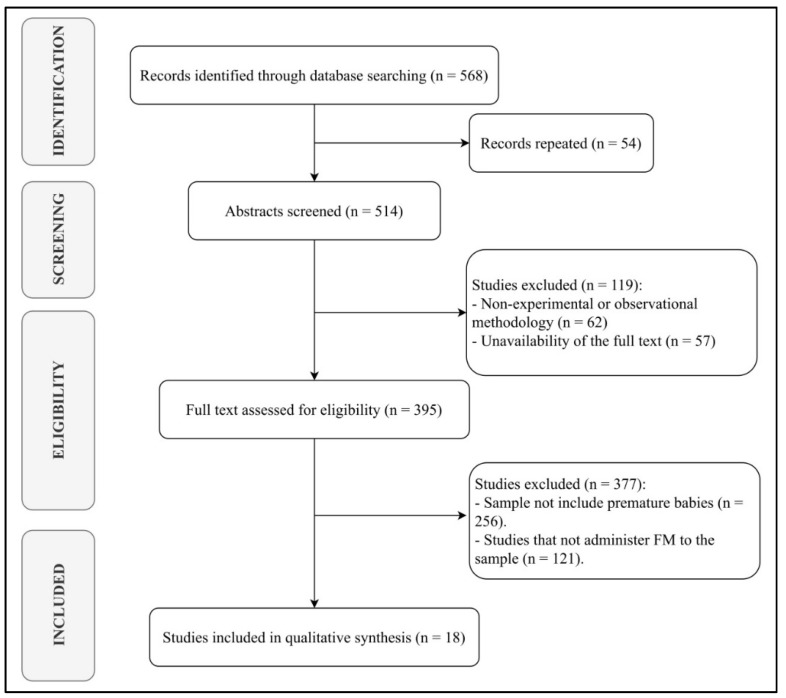
PRISMA flow diagram.

**Table 1 children-09-00150-t001:** Search strategy according to the focused question (PICO).

Database	Search Equation
PubMed	Infant Formula [Mesh] AND “Infant, Premature” [Mesh]
Medline	(MH “Infant Formula”) AND (MH “Infant, Premature”)
Cinahl	(MH “Infant Formula”) AND (MH “Infant, Premature”)
Web of Science	TOPIC: (“infant formula”) AND TOPIC: (premature)
ClinicalTrials	Infant Formula AND Premature Infant
Scopus	(TITLE-ABS-KEY (“infant formula”) AND TITLE-ABS-KEY (premature))

**Table 2 children-09-00150-t002:** Characteristics of the interventions of the studies analyzed.

Authors	Intervention		Time of Intervention	Caloric Density	Sample Characteristics
	Experimental Group	Control Group			
Baldassarre et al. (2019) [19]	Intact protein FM	Highly hydrolysed FM	14 first days of life	0.008 kcal/100 mL	Very preterm
Costa et al. (2018) [20]	FM	DHM	Up to 36 weeks GA or discharge from hospital	FM: 3.5 g protein/100 kcal DHM: 2.5 g protein/100 kcal	Very preterm
Toftlund et al. (2018) [21]	Group 1: FM Group 2: DHM	BM	Until four months or hospital discharge	FM: 68 kcal/100 mL DHM: 17.5 kcal/5 paquetes	Very preterm and extremely preterm
Corpeleijn et al. (2016) [22]	FM	DHM	Ten days	Not described	Very low birth weight preterm
O’Connor et al. (2016) [23]	FM	DHM	Until three months or hospital discharge	FM: 67–80 kcal/100 mL and 3 g de proteína/100 kcal	Very low birth weight preterm
Willteitner et al. (2017) [24]	Concentrated liquid fortifier	Powder fortifier	Until tolerance is reached for three consecutive days	80 kcal/100 mL	Very low birth weight preterm
Da Cunha et al. (2016) [25]	BM and FM	BM	4–6 months	FM: increment of 20 kcal/día	Very low birth weight preterm
Kim et al. (2015) [26]	Liquid fortifier	Powder fortifier	29 days	Liquid fortifier: 3.6 g protein/100 kcal Powder fortifier: 3 g protein/100 mL	Very low birth weight preterm
Chen et al. (2020) [27]	FM	BM	28 days	Not described	Moderate preterm
Martins-Celini et al. (2018) [28]	Group 1: FM Group 2: FM and BM	BM	Until hospital discharge	Not described	Very low birth weight preterm
Brownell et al. (2018) [29]	Group 1: FM Group 2: DHM	BM	Up to 36 weeks postmenstrual age or discharge from hospital	67 kcal/100 mL	Very preterm
Pillai et al. (2018) [30]	Liquid fortifier	*---*	Until hospital discharge	80 kcal/100 mL	Preterm
Kim et al. (2017) [31]	FM	DHM	Until 130 mL/kg/day	FM: 80 kcal/100 mL	Very premature and very low birth weight
Lofti et al. (2016) [32]	BM and FM	FM	*Not described*	Not described	Very low birth weight preterm
Fernandes et al. (2019) [33]	BM and FM	BM	12–15 months	73 kcal/100 mL	Very low birth weight preterm
Hogewind-Schoonenboom et al. [34]	Group 1: 0–57% BM and 100–43% FM Group 2: 58–96% BM and 42–4% FM	97–100% BM and 3–0% FM	Until 120 mL/kg/day or 28 days is reached	Not described	Very preterm
Jang et al. (2018) [35]	FM in infants with food intolerance	FM in healthy infants	Until hospital discharge	Not described	Very preterm

FM: formula milk; BM: Breast milk; DHM: Donated human milk.

**Table 3 children-09-00150-t003:** Methodological characteristics of the studies analyzed.

Authors	Design	Sample Size	Inclussion Criteria	Exclussion Criteria	JADAD Scale				LE
					RD *	BD **	WD ***	FS	
Baldassarre et al. (2019) [19]	RCT	60	28–33 weeks GA. Birth of a singleton or twins. Birth weight between 700–1750 g and appropriate for GA. Enteral intake less than 30 mL/kg/day or none at baseline.	Apgar less than 4 at five minutes of life. Presence of chronic diseases, metabolic disturbance, congenital malformation, unstable blood pressure, and/or intraventricular haemorrhage. History of surgery. Need for ventilator or more than 40% inspired oxygen fraction.	2	2	1	5	I
Costa et al. (2018) [20]	QES	70	GA less than 32 weeks. Beginning of enteral feeding in the first seven days of life. Breast feeding not available or insufficient	Presence of infections, congenital malformation, abnormal prenatal Doppler flow velocity and/or abnormal prenatal velocimetry.	2	0	1	3	II
Toftlund et al. (2018) [21]	RCT	235	GA less than 32 weeks.	Serious illnesses or circumstances influencing feeding.	1	0	1	2	I
Corpeleijn et al. (2016) [22]	RCT	373	Birth weight less than 1500 g.	Toxic substance use during pregnancy. Presence of congenital defects or anomalies and/or infections. History of perinatal asphyxia with umbilical pH below 7 and/or intake of formula milk prior to surgery.	2	2	1	5	I
O’Connor et al. (2016) [23]	RCT	363	Birth weight less than 1500 g. Start of enteral nutrition in the first seven days of life.	Presence of congenital defects or anomalies. History of severe birth asphyxia.	2	2	1	5	I
Willteitner et al. (2017) [24]	RCT	70	Birth weight between 500 and 1499 g.	Presence of congenital anomalies and/or gastrointestinal diseases.	2	2	1	5	I
Da Cunha et al. (2016) [25]	RCT	53	Birth weight less than 1500 g. Infants admitted to the Neonatal Intensive Care Unit.	Presence of malformations, hydrocephalus, chromosomal abnormalities, hydrops fetalis, infections and/or necrotising enterocolitis. Born of a twin pregnancy. Consumption of toxic substances and/or corticosteroids during pregnancy.	2	0	1	3	I
Kim et al. (2015) [26]	RCT	129	GA less than 33 weeks. Birth weight between 700 and 1500 g. Enteral feeding in the Neonatal Intensive Care Unit and during the first 21 days of life.	Apgar less than 5 at five minutes of life. Presence of congenital anomalies. History of severe intraventricular haemorrhage, major abdominal surgery, severe asphyxia, necrotising enterocolitis, and/or consumption of probiotics and/or postnatal corticosteroids.	2	0	0	2	I
Chen et al. (2020) [27]	PCS	60	GA less than 36 weeks and/or birth weight less than 2500 g. Admission to Neonatal Intensive Care Unit.	Presence of congenital malformations.	0	0	0	0	II
Martins-Celini et al. (2018) [28]	RCS	649	Birth weight less than 1500 g.	Presence of congenital malformations.	0	0	0	0	II
Pillai et al. (2018) [30]	PCS	29	None.	Presence of congenital and/or macrosomic anomalies. Diagnosis of multi-organ and/or intestinal dysfunction.	0	0	1	1	II
Kim et al. (2017) [31]	RCS	90	Birth weight less than 1500 g. GA less than 32 weeks. Admission to the Neonatal Intensive Care Unit.	Presence of congenital and/or metabolic abnormalities. BM-fed exclusive before 130 mL/kg/day.	0	0	1	1	II
Lofti et al. (2016) [32]	PCS	58	Birth weight less than 1500 g. Admission to Neonatal Intensive Care Unit.	None.	0	0	0	0	II
Fernandes et al. (2019) [33]	PCS	51	GA less than 37 weeks. Birth weight less than 1250 g.	None.	0	0	0	0	II
Hogewind-Schoonenboom et al. [34]	RCS	174	GA less than 32 weeks. Birth weight less than 1750 g.	None.	0	0	0	0	II
Jang et al. (2018) [35]	RCS	60	GA between 29 and 32 weeks. Admitted to the Neonatal Intensive Care Unit.	Presence and/or history of sepsis, necrotising enterocolitis and/or asphyxia.	0	0	0	0	II

GA: Gestational age; LE: Level of evidence; PCS: Prospective cohort study; RCS: Retrospective cohort study. * RD: Randomization (1 point if randomization is mentioned; 2 points if the method of randomization is appropriate). ** BD: Blinding (1 point if blinding is mentioned; 2 points if the method of blinding is appropriate). *** WD: Withdrawals (1 point if the number and reasons in each group are stated).

## Data Availability

The data presented in this study are available on request from the corresponding author.

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
