# Peer review of "Effects of Formula Milk Feeding in Premature Infants: A Systematic Review"

_children, 2022, doi:10.3390/children9020150_

Round 1

Reviewer 1 Report

The review must be redone in a more schematic and understandable way and based on data. Opinions and extrapolations of a personal nature must be abolished.

The subject is interesting but requires major changes. The impression is that the review starts from pre-established and preconceived conclusions and that is that FM is similar or superior to donated human milk for supplementation with breast milk. There is a substantial difference between the different categories of premature babies and this aspect is not really considered here. Then there are statements that should be the result of a serious scientific comparison extrapolated from the studies but often the applied methodology is not this.

Author Response

Dear Editor and Reviewer #1 of Children:

Thank you very much for your suggestions and contributions to improve the quality of the manuscript. Following your indications, we respond, point by point, to the reviewers' comments.

In the text, all the modified or added sentences have been written in red to facilitate the correction by the reviewers.

  1. The review must be redone in a more schematic and understandable way and based on data. Opinions and extrapolations of a personal nature must be abolished.

The subject is interesting but requires major changes. The impression is that the review starts from pre-established and preconceived conclusions and that is that FM is similar or superior to donated human milk for supplementation with breast milk. There is a substantial difference between the different categories of premature babies and this aspect is not really considered here. Then there are statements that should be the result of a serious scientific comparison extrapolated from the studies but often the applied methodology is not this.

The entire manuscript has been thoroughly revised on your advice.

  1. The abstract is poorly done. It already starts with a personal hypothesis. The conclusion is tendentious and is not evident from the review.

The authors have rewritten the Abstract in response to your correction.

  1. The introduction makes no mention of the fact that WHO guidelines recommend fortified donor's milk, as an alternative to fortified mother’s milk and, in the absence of donated human milk, the formula for premature babies.

The authors have added this detail in the Introduction.

  1. 124 to 125 should be shifted into materials and methods.

The authors do not understand this correction: Tables 2 and 3, in setting out the characteristics of the studies selected as Results, we consider that they belong in that section.

  1. The results paragraph is confusing, generic with a poor representation of the numerical data of the studies analyzed in the review

A total of 18 articles were selected, of which eight were experimental studies [19–26] 117 and the remaining ten were observational studies [27–36] (Figure 1). The methodological 118 characteristics of the analysed studies are detailed in Table S1 and the characteristics of 119 the applied interventions are presented in Table 2. 120 The results must be represented on the basis of this distinction.

The results section is long and not very detailed. The results are not understandable. The results should be summarized both in the text and in a table.

In the text they should be divided by theme: Auxological data, Incidence of complications, Microbiota, Tolerance, Long-term follow up

The authors have modified the presentation of the Results following your advice.

  1. The discussion is full of opinions unsupported by data or scientific references and often contradicting the text.

After the analysis of the obtained results, it can be confirmed that FM does cause different effects in neonates compared to BM or DHM.

It is a generic phrase, which expresses a personal opinion and which is not inferred from the data of the text. Confirm with respect to what?

The indicated sentence has been replaced by one that responds to the objective of the research in a more modest and appropriate way.

  1. The FM had a similar tolerance to human milk (either BM or DHM) except in two studies [26,34]. In one study, this result could be associated with the presence of casein in the composition of FM [26], although, in the other case, no data are provided to justify this finding.

Are you sure of this simplification?

The authors have rewritten the sentence to be more cautious about this hypothesis. In addition, we have added a bibliographic reference that supports that this could be a plausible explanation for this finding.

  1. A tendency for NEC, sepsis and bronchopulmonary dysplasia was found when FM [23,31,32] or a powdered fortifier [24] was used.

The two things must be distinguished and if associated the type of association must be specified.

The authors have expanded and improved the explanation of this finding.

  1. Psychomotor development was not affected by the feeding method, although the highest scores were found for feeding BM supplemented with FM [23,25]. However, exclusive DHM [23] or BM [25] feeding resulted in a higher incidence of neurodevelopmental delay, possibly due to the lack of supplementation.

Were DHM or BM fortified? Does this seem a minor detail to you?

No, we do not consider it a minor detail at all. As it is a very relevant aspect whenever fortifier is added to breastmilk or donated human milk, the authors have referred to this detail.

In this case, the studies mentioned do not indicate that any fortifier was added to breast milk or donated human milk.

The authors have specified this in the text.

  1. You don't find it contradictory to say before that Psychomotor development was not affected by the feeding method” and then “exclusive DHM [23] or BM [25] feeding resulted in a higher incidence of neurodevelopmental delay”

The authors acknowledge the error.

We wanted to indicate that there were no statistically significant differences in psychomotor development. However, in these two articles we did find higher incidence results (without being statistically significant in the intergroup comparison).

We, the authors, have specified this detail in the text.

  1. Morbidity did not differ with respect to the type of feeding [19,22,24,26,34,35], although in several studies there was a higher incidence with FM [30–32].

How can you say that there is no difference if you then say that in many studies there is a higher incidence?

In this case, the authors also wanted to make reference to the fact that the results were higher in all cases without obtaining statistically different results when comparing the groups.

We, the authors, have specified this detail in the text.

  1. Regarding short- and long-term benefits, studies comparing the effects of feeding during hospital stay [19,20,22,24,26–32,34–36], showed that there were no differences in tolerance or duration of admission regardless of the type of feeding [19,20,22,24,26], although, in one study, FM prolonged the time to reach full enteral feeding and the duration of the hospital stay [31].

You cannot write there is no difference if you then highlight the differences

The authors have rewritten the phrase indicated so as not to contradict ourselves.

  1. Regarding the methodological limitations of this review, it should be noted that many studies are observational, in which the lack of randomisation and control group limits the generalisability of the results obtained. It should also be noted that the sample sizes were small, thus the results may not be entirely accurate and extrapolation to the entire population of preterm infants is not possible. However, as strengths, it is worth highlighting that the methods used to measure the different variables are objective. In addition, this work has included a large number of studies, covering all degrees of prematurity and the application of the different types of feeding that are currently available.

All the more reason, on the basis of all these limitations, conclusive personal judgments which are misleading must be abolished.

The conclusion is totally inconsistent with the text and with the bibliography.

On the other hand, FM provides a similar quality and it is even more beneficial in some aspects than DHM, as DHM does not seem to meet the nutritional objectives of BM.

It is a personal judgment not corroborated by the data. If you evaluate the comparison between FM and DHM does it include fortification or not? It is now well established that unfortified DHM does not meet the nutritional needs of VLBW infants and that human milk must be fortified and the cut off of the human milk volume to be fortified has dropped to 60-80 mg / kg.

The authors have rewritten and shortened the Conclusions to convey only factual information based on this research.

Once again, thank you very much for the time spent and the interest shown in this work; as well as in the positive evaluations you have given of it.

Receive a warm greeting,

The authors.

Reviewer 2 Report

This paper seems to evaluate the evidence for using infant formula among preterm infants. It is important to understand more clearly the use cases for infant formula and how it can help to promote optimal outcomes in preterm infants, and a clear synthesis of these research findings is needed. The authors' work is promising, and stating a clear research question up front will strengthen this work. Currently, the lack of a clear research question leads to some challenges in understanding the selection of studies and interpretation of the findings. I encourage the authors to state their research question(s) explicitly, so that it is clear which outcome(s) are being used to judge the efficacy of formula compared to human milk alone in preterm infants. Once that is clear, it would help to organize the results and discussion by outcome type rather than by study. This will help the reader evaluate the evidence for each outcome in order to determine the best use cases for formula among preterm infants, and to what degree formula vs human milk might be used. More specific comments and suggestions are included below.

Abstract

  • The population of interest should be clear from the outset, i.e., in the first sentence.
  • The first sentence and penultimate sentence seem repetitive. Please specify why formula is a good nutritional option for this population in particular.
  • If formula is more beneficial than donor milk, please qualify why you are stating this. What metric are you using as the basis of this statement? In the results you state that some adverse outcomes (death, sepsis, NEC, bronchopulmonary dysplasia) were lower in the DHM group.

Introduction

  • Is human milk nutritionally insufficient for all infants in the NICU (line 50)? If so, why? If not, please qualify. It seems this should be important to set up your research question and subsequently your argument.
  • Do all mothers demonstrate an inability to offer human milk (lines 55-7)?
  • I do not understand the final paragraph of the introduction. Will you please state your research question explicitly? Lines 57-60 suggest you are interested in a comparative study of human milk vs formula milk, yet then you state you wanted to “identify the effects of FM feeding.” This is confusing.
  • What are you trying to say in lines 62-4? What are the relevant previous findings?

Methods

  • The first sentence of the final paragraph (line 107) seems to be missing a verb.

Results

  • It would be helpful if Table 2 included the sample sizes and outcomes of interest, i.e., the outcomes you specified in the methods, as this would help readers understand how you made your final assessments. It would also be great to include the findings related to the outcomes, but at minimum the outcomes should be listed.
  • In your methods, you state that you were interested in a FM intervention with human milk as a control (lines 70-1), yet you included studies that investigated the effects of formula milk composition and/or fortifiers (line 127). Please explain why.
  • The presentation of the results is confusing to me. For example, would help if the results were organized by specific outcome and described how many papers had results in favor of FM, in favor of BM/DHM, and how many found no difference, and what the populations, sample sizes, and observed effect sizes were in each of those studies. For example, it seemed strange to read a sentence about weight in the middle of a paragraph that is mostly about the gut microbiome. Even if some studies report on multiple outcomes, grouping by outcome rather than study could help the reader stay focused on each specific outcome as you address it.
    • The sentences in lines 129-133 seem contradictory. What “they” were not affected by FM intake?
    • When discussing fortification, please be clear about what the numbers refer to, e.g., does 67 kcal per 100 mL refer to what was added or the final caloric density?
    • Please be consistent about referring to authors by name in the results. I find it is helpful (for my own comprehension) when author names are used rather than not. For example, in the paragraph on microbiota (lines 150-178), please refer to each study specifically. You mention that one study found differences in alpha- and beta-diversity by feeding mode (lines 153-5), but then in lines 165-6, there is a contradictory finding in a different study. Using author names here would really help the reader follow to which study you are referring.
  • Please be sure to define all acronyms at first use. What is “MF” in line 135? What is “LHD” in line 140?
  • Your research aim seems to be about feeding mode, but you go into detail about other factors such as delivery type and maternal factors (e.g., preeclampsia and GDM) (lines 154-8 & 182-5), and you have an entire section on outcomes based on feeding intolerance. I thought feeding intolerance was an outcome of interest and do not understand the relevance of it as a predictor here. Was this part of your research question?
  • What objective are you referring to in lines 199-202? This is a standalone paragraph, and it is unclear to what you are referring.
  • If comparing fortification is in fact a central component of your research question, then please explain what are the important differences between the standard powder and liquid concentrate described in lines 203-210.

Discussion

  • Referring to the studies by first author name in the results and then again the discussion, could help the reader follow your train of thought more closely. Right now it is difficult to match what I read in the results with what I read in the discussion.
  • You state that the findings of lower HC among neonates who received DHM were congruent with another study, but the other study was measuring BM intake (lines 281-4). Are you implying that DHM and BM are congruent? If so, please qualify this.
  • Is it sound to conclude that psychomotor development was not affected by feeding mode, with only two studies reporting these outcomes (lines 285-6)?
  • What about higher neurodevelopmental delay (286-8)? Why might the lack of supplementation result in delays? Is this based solely on the fact that neonates did not receive supplementation?
  • Lines 291-4 do not make sense to me. If not providing BM or DHM implied higher Enterobacteriaceae, why is DHM pasteurization relevant?
  • You state that FM does not cause changes in microbial diversity, possibly due to antibiotics (line 298). This currently reads as a non sequitur. What do antibiotics have to do with FM?
  • You state that morbidity did not differ by feeding mode (lines 309-310). In the discussion you list six studies, but you only reported on two in the second-to-last paragraph of the results, and one study did find differences. Please clarify.
  • When discussing growth outcomes, please be clear about the timelines you are referring to: short- vs long-term growth, and within that, what the timeframes are.

Conclusion

  • Please be clear about which criteria you are using to evaluate FM vs DHM.
  • What do you mean by “nutritional objectives” of BM?

Author Response

Dear Editor and Reviewer #2 of Children:

Thank you very much for your suggestions and contributions to improve the quality of the manuscript. Following your indications, we respond, point by point, to the reviewers' comments.

In the text, all the modified or added sentences have been written in red to facilitate the correction by the reviewers.

  1. Abstract: The population of interest should be clear from the outset, i.e., in the first sentence.

The authors have added an opening sentence referring to the study population.

  1. Abstract: The first sentence and penultimate sentence seem repetitive. Please specify why formula is a good nutritional option for this population in particular.

The authors have rewritten the Abstract following your advice.

  1. Abstract: If formula is more beneficial than donor milk, please qualify why you are stating this. What metric are you using as the basis of this statement? In the results you state that some adverse outcomes (death, sepsis, NEC, bronchopulmonary dysplasia) were lower in the DHM group.

The authors have removed this sentence because it was not faithful to the results obtained and their generalizability.

  1. Introduction: Is human milk nutritionally insufficient for all infants in the NICU (line 50)? If so, why? If not, please qualify. It seems this should be important to set up your research question and subsequently your argument.

The authors have rewritten the phrase.

  1. Introduction: Do all mothers demonstrate an inability to offer human milk (lines 55-57)?

The authors have rewritten the sentence.

  1. Introduction: I do not understand the final paragraph of the introduction. Will you please state your research question explicitly? Lines 57-60 suggest you are interested in a comparative study of human milk vs formula milk, yet then you state you wanted to “identify the effects of FM feeding.” This is confusing.

The authors have rewritten the two parts of the Introduction to make them consistent.

  1. Introduction: What are you trying to say in lines 62-64? What are the relevant previous findings?

The authors have rewritten the previous hypothesis.

  1. Methods: The first sentence of the final paragraph (line 107) seems to be missing a verb.

Yes, the authors have corrected the sentence.

  1. Results: It would be helpful if Table 2 included the sample sizes and outcomes of interest, i.e., the outcomes you specified in the methods, as this would help readers understand how you made your final assessments.

For reasons of available space, the sample sizes and other relevant information are described in Table S1.

  1. Results: The presentation of the results is confusing to me. For example, would help if the results were organized by specific outcome and described how many papers had results in favor of FM, in favor of BM/DHM, and how many found no difference, and what the populations, sample sizes, and observed effect sizes were in each of those studies. For example, it seemed strange to read a sentence about weight in the middle of a paragraph that is mostly about the gut microbiome. Even if some studies report on multiple outcomes, grouping by outcome rather than study could help the reader stay focused on each specific outcome as you address it.

The Results section has been rewritten and heavily edited to improve the communication of findings.

  1. Results: The sentences in lines 129-133 seem contradictory. What “they” were not affected by FM intake?

The authors wanted to refer to weight gain and head circumference. The sentence has been rewritten.

  1. Results: When discussing fortification, please be clear about what the numbers refer to, e.g., does 67 kcal per 100 mL refer to what was added or the final caloric density?

The authors have specified this detail.

  1. Results: Please be consistent about referring to authors by name in the results. I find it is helpful (for my own comprehension) when author names are used rather than not. For example, in the paragraph on microbiota (lines 150-178), please refer to each study specifically. You mention that one study found differences in alpha- and beta-diversity by feeding mode (lines 153-5), but then in lines 165-6, there is a contradictory finding in a different study. Using author names here would really help the reader follow to which study you are referring.

The authors have specified this detail.

  1. Results: Please be sure to define all acronyms at first use. What is “MF” in line 135? What is “LHD” in line 140?

The authors greatly regret the existence of errata in the document.

The entire text has been thoroughly revised to correct all such errors.

  1. Results: Your research aim seems to be about feeding mode, but you go into detail about other factors such as delivery type and maternal factors (e.g., preeclampsia and GDM) (lines 154-8 & 182-5), and you have an entire section on outcomes based on feeding intolerance. I thought feeding intolerance was an outcome of interest and do not understand the relevance of it as a predictor here. Was this part of your research question?

The authors acknowledge the error and have removed from the text the references to those issues that are not relevant to the purpose of the research.

  1. Results: What objective are you referring to in lines 199-202? This is a standalone paragraph, and it is unclear to what you are referring.

The paragraph has been rewritten and moved in the text to make it congruent with adjoining paragraphs.

  1. Discussion: Referring to the studies by first author name in the results and then again the discussion, could help the reader follow your train of thought more closely. Right now it is difficult to match what I read in the results with what I read in the discussion.

The authors have added the names of the authors of the studies on more occasions.

  1. Discussion: You state that the findings of lower HC among neonates who received DHM were congruent with another study, but the other study was measuring BM intake (lines 281-284). Are you implying that DHM and BM are congruent? If so, please qualify this.

The authors acknowledge that this statement was incorrect and have removed it from the Discussion.

  1. Discussion: Is it sound to conclude that psychomotor development was not affected by feeding mode, with only two studies reporting these outcomes (lines 285-286)?

The authors have limited the degree of affirmation to not significantly affected in the two studies analysed.

  1. Discussion: You state that morbidity did not differ by feeding mode (lines 309-310). In the discussion you list six studies, but you only reported on two in the second-to-last paragraph of the results, and one study did find differences. Please clarify.

The authors have limited the degree of affirmation to not significantly affected in the studies referenced.

  1. Discussion: When discussing growth outcomes, please be clear about the timelines you are referring to: short- vs long-term growth, and within that, what the timeframes are.

The authors have specified the time periods to which the research analysed refers.

  1. Conclusion: Please be clear about which criteria you are using to evaluate FM vs DHM.

The comparison with Donated Human Milk has been removed from the Conclusions because it is not the purpose of this research to respond to such a comparison.

Once again, thank you very much for the time spent and the interest shown in this work; as well as in the positive evaluations you have given of it.

Receive a warm greeting,

The authors.

Round 2

Reviewer 1 Report

The text has improved a lot and the addition of paragraphs has been excellent. Now the criticality that has remained concerns the understanding of the description of the results of the studies analyzed both in the results section and in the discussion section. We often talk about groups being compared but it is not clear which ones. They should be better specified. Then there is often no consistency between one comparison and another. The paragraphs show the results of comparisons with very different objectives that should be better categorized. A table would improve understanding.

Example

If you talk about the long-term effects, it is good to report the results starting from the remote effect that has been studied. Taken all together, it is very difficult to understand how a variable has affected the goal, which in this case is the long-term effect.

The entity of the analyzed sample is never indicated. When it is useful it is worth indicating whether the sample analyzed is large or small. This gives meaning to the results you are reporting.

  1. the rate of weight gain decreased by 0.17 g/kg/day and head circumference (HC) growth also decreased.

Decreased: compared to what?

  1. reduced (non-significantly) the incidence of neuropsychomotor development?????
  2. 33.3% of infants fed exclusively BM with impaired development and only 28% of infants fed BM and the supplement: this sentence is incomprehensible

226/227 The results indicated that intolerance occurred in 14% of the infants, 3% had late onset sepsis  after 28 days of fortification and 7% had ventilator-associated pneumonia. Thus reported these data have no meaning. Is there a comparison, a specific evaluation?

257 in the faeces: whose?

267 in both groups: which groups?

296 the groups: Which? It must be specified

301-304. In the long term, Willeitner et al. [24] found that a 30 kcal/oz liquid preterm formula  for fortifying human milk at a caloric density of 24 kcal/oz did not result in significant  improvement in weight, feeding tolerance, caloric intake, sepsis or mortality compared to  a standard fortifier (whose composition and caloric density are not specified).

Here another comparison, different from the previous one, has been inserted and then we go back, I think, to the previous comparison. It must be specified, it is not clear

307 was lower in the DHM group. Which comparison are you referring to? Was DHM included in the previous ones?

320-321. Complete enteral feeding was achieved after longer periods with FM [31] or when using liquid fortifier [30] at 31 and 30 days, respectively. 30 e 31? Longer than what?

The conclusion is a bit concise but it could be fine.

Author Response

Dear Editor and Reviewer #1 of Children:

Thank you very much for your suggestions and contributions to improve the quality of the manuscript. Following your indications, we respond, point by point, to the reviewers' comments.

In the text, all the modified or added sentences have been written in red to facilitate the correction by the reviewers.

  1. The entity of the analyzed sample is never indicated. When it is useful it is worth indicating whether the sample analyzed is large or small. This gives meaning to the results you are reporting.

The authors have added a table with the sample sizes, the characteristics of the sample used in each investigation and the risk of bias for each of them.

  1. The rate of weight gain decreased by 0.17 g/kg/day and head circumference (HC) growth also decreased. Decreased: compared to what?

The authors have rewritten the sentence to facilitate its interpretation.

  1. Reduced (non-significantly) the incidence of neuropsychomotor development???? 33.3% of infants fed exclusively BM with impaired development and only 28% of infants fed BM and the supplement: this sentence is incomprehensible

The authors have rewritten the sentence to facilitate its interpretation.

  1. The results indicated that intolerance occurred in 14% of the infants, 3% had late onset sepsis after 28 days of fortification and 7% had ventilator-associated pneumonia. Thus reported these data have no meaning. Is there a comparison, a specific evaluation?

The authors have rewritten the sentence to facilitate its interpretation.

  1. 257: in the faeces: whose?

The authors have rewritten the sentence to facilitate its interpretation.

  1. 296: the groups: Which? It must be specified

The authors have rewritten the sentence to facilitate its interpretation.

  1. 301-304: In the long term, Willeitner et al. [24] found that a 30 kcal/oz liquid preterm formula for fortifying human milk at a caloric density of 24 kcal/oz did not result in significant improvement in weight, feeding tolerance, caloric intake, sepsis or mortality compared to a standard fortifier (whose composition and caloric density are not specified). Here another comparison, different from the previous one, has been inserted and then we go back, I think, to the previous comparison. It must be specified, it is not clear.

The authors have rewritten the sentence to facilitate its interpretation.

  1. 307: was lower in the DHM group. Which comparison are you referring to? Was DHM included in the previous ones?

The authors have rewritten the sentence to facilitate its interpretation.

  1. 320-321: Complete enteral feeding was achieved after longer periods with FM [31] or when using liquid fortifier [30] at 31 and 30 days, respectively. 30 e 31? Longer than what?

The authors have rewritten the sentence to facilitate its interpretation.

  1. The conclusion is a bit concise but it could be fine.

The Conclusions have been slightly expanded to be more self-explanatory.

Once again, thank you very much for the time spent and the interest shown in this work; as well as in the positive evaluations you have given of it.

Receive a warm greeting,

The authors.

Reviewer 2 Report

Thank you to the authors for their thoughtful revisions. The manuscript is greatly improved. This study represents an important synthesis of research on the use of formula milk in preterm infants. There are still some areas of the paper that would benefit from more careful explanation or clarification, including the authors objective(s). The authors postulate that not all formulations have the same effects on preterm infants, but there is no synthesis of the findings in relation to formulation. Is this actually within the scope of the present paper? Please see more detailed comments below.

The abstract, introduction, and methods are greatly improved.

Results:

This section is much more organized. There are still some areas that could be streamlined and/or clarified. Even with better organization, some of the similar outcome measures still seem to be reported in different places. How does this placement support the authors' argument? 

  • The section labeled "tolerance" includes a couple of paragraphs at the end about tolerance specifically, and it also includes growth outcomes. Are these one and the same? Perhaps it would help for the authors to define "tolerance."
  • It seems to me that it would be beneficial to enumerate anthropometric outcomes together, tolerance parameters together (do these include anthropometrics?), neurodevelopment, time to enteral feeding, etc. Is there a rationale for presenting it differently? For example, it might be helpful for readers to describe how many studies found differences in weight, length & HC gains and how many did not.
  • The authors open the first section with "the reviewed research identified..." disparities in weight gains and HC. Is this a synthesis of all studies, or do only some studies report these findings? This is unclear.
  • Lines 200-3 seem like these findings might fit more within the "long-term outcomes" section.
  • Beginning line 204, the authors state that other studies identified non-significant reductions in neuropsychomotor development. I see only one study referenced. Were there others? Is neuropsychomotor development a measure of tolerance?
  • There are still some seeming inconsistencies. Lines 190-1: the authors state weight gain & HC were not affected by FM, and then they state weight increased significantly. Please clarify.
  • Lines 238-240 read as if a single study is being reported, yet two different studies are included. Please clarify.
  • Lines 277-283: are the findings by intolerance (measured how?) the same for infants receiving BM & FM? Please clarify.
  • Lines 290-2: how germane is it to the discussion that SGA infants had greater long-term gains than AGA infants? Would you expect this to differ? Please clarify.
  • Long-term follow up: please define long-term in the results (or methods) vs the discussion.
  • Lines 296-9: how does intake relate to the incidence of complications? Lines 298-9: by what feeding modality was hospitalization longer?
  • Line 301: define long-term. Does this belong in the incidence of complications section, or the long-term section?

Discussion:

  • The characteristic signs of tolerance problems should be defined somewhere up front (results, perhaps) rather than waiting until the discussion.
  • What were the most important findings related to tolerance and complete enteral feeding (and the other outcomes you describe below)? If you can describe tolerance somewhere else, then in your discussion you can lead with what the most common findings were (as you do in lines 373-374). Also, think about comparing things like population, sample size, methodology/measurement, and effect size when comparing the findings in the discussion. This will bolster your final argument.
  • Lines 373-7: define what you mean by growth. 

Conclusion:

  • Does FM improve neurological development? I noticed the non-significant findings but may have missed other significant findings. How strong were the effect sizes, and what were the sample sizes? Reorganizing to report all data related to this outcome together could provide better support for this argument.
  • It may be worth mentioning the beneficial effects of the BM/FM combination on the gut microbiome. There could be some longer-term health implications.

Author Response

Dear Editor and Reviewer #2 of Children:

Thank you very much for your suggestions and contributions to improve the quality of the manuscript. Following your indications, we respond, point by point, to the reviewers' comments.

In the text, all the modified or added sentences have been written in red to facilitate the correction by the reviewers.

  1. Results: The section labeled "tolerance" includes a couple of paragraphs at the end about tolerance specifically, and it also includes growth outcomes. Are these one and the same? Perhaps it would help for the authors to define "tolerance".

It seems to me that it would be beneficial to enumerate anthropometric outcomes together, tolerance parameters together (do these include anthropometrics?), neurodevelopment, time to enteral feeding, etc. Is there a rationale for presenting it differently?

The authors have added a sentence to justify presenting the data on tolerance and growth together. In addition, we have modified the name of this Results subsection.

  1. The authors open the first section with "the reviewed research identified..." disparities in weight gains and HC. Is this a synthesis of all studies, or do only some studies report these findings? This is unclear.

The authors have added the bibliographic reference that conveys this information.

  1. Lines 200-3 seem like these findings might fit more within the "long-term outcomes" section.

The authors have moved the indicated sentences to the subsection “Long-term follow up”.

  1. Beginning line 204, the authors state that other studies identified non-significant reductions in neuropsychomotor development. I see only one study referenced. Were there others? Is neuropsychomotor development a measure of tolerance?

The authors have rewritten the sentences indicated and added the bibliographic references.

  1. There are still some seeming inconsistencies. Lines 190-191: the authors state weight gain & HC were not affected by FM, and then they state weight increased significantly. Please clarify.

The authors have rewritten the sentences indicated.

  1. Lines 238-240: read as if a single study is being reported, yet two different studies are included. Please clarify.

The authors have rewritten the sentences indicated.

  1. Lines 277-283: are the findings by intolerance (measured how?) the same for infants receiving BM & FM? Please clarify.

The authors have rewritten the sentences indicated.

  1. Long-term follow up: please define long-term in the results (or methods) vs the discussion.

The authors have added this detail next to the title of the subsection.

  1. Lines 298-299: by what feeding modality was hospitalization longer?

The authors have rewritten the sentence to improve its interpretation.

  1. Discussion: The characteristic signs of tolerance problems should be defined somewhere up front (results, perhaps) rather than waiting until the discussion.

This detail has been added in the Results section when this topic is discussed.

  1. What were the most important findings related to tolerance and complete enteral feeding (and the other outcomes you describe below)? If you can describe tolerance somewhere else, then in your discussion you can lead with what the most common findings were (as you do in lines 373-374). Also, think about comparing things like population, sample size, methodology/measurement, and effect size when comparing the findings in the discussion. This will bolster your final argument.

The authors have rewritten several paragraphs of the Discussion to improve the exposition of the arguments discussed.

  1. Lines 373-377: define what you mean by growth.

The authors have added the information you indicated.

  1. Conclusion: Does FM improve neurological development? I noticed the non-significant findings but may have missed other significant findings. Reorganizing to report all data related to this outcome together could provide better support for this argument.

It may be worth mentioning the beneficial effects of the BM/FM combination on the gut microbiome. There could be some longer-term health implications.

The authors have rewritten and expanded the Conclusions in response to your comments.

Once again, thank you very much for the time spent and the interest shown in this work; as well as in the positive evaluations you have given of it.

Receive a warm greeting,

The authors.